# Impact of Vitamin D on Immunopathology of Hashimoto’s Thyroiditis: From Theory to Practice

**DOI:** 10.3390/nu15143174

**Published:** 2023-07-17

**Authors:** Filip Lebiedziński, Katarzyna Aleksandra Lisowska

**Affiliations:** Department of Physiopathology, Medical University of Gdańsk, 80-211 Gdansk, Poland; flebiedzinski@gumed.edu.pl

**Keywords:** Hashimoto’s thyroiditis, vitamin D, autoimmunity, cytokines, anti-thyroid antibodies

## Abstract

Hashimoto’s thyroiditis (HT) is a common autoimmune disease affecting the thyroid gland, characterized by lymphocytic infiltration, damage to thyroid cells, and hypothyroidism, and often requires lifetime treatment with levothyroxine. The disease has a complex etiology, with genetic and environmental factors contributing to its development. Vitamin D deficiency has been linked to a higher prevalence of thyroid autoimmunity in certain populations, including children, adolescents, and obese individuals. Moreover, vitamin D supplementation has shown promise in reducing antithyroid antibody levels, improving thyroid function, and improving other markers of autoimmunity, such as cytokines, e.g., IP10, TNF-α, and IL-10, and the ratio of T-cell subsets, such as Th17 and Tr1. Studies suggest that by impacting various immunological mechanisms, vitamin D may help control autoimmunity and improve thyroid function and, potentially, clinical outcomes of HT patients. The article discusses the potential impact of vitamin D on various immune pathways in HT. Overall, current evidence supports the potential role of vitamin D in the prevention and management of HT, although further studies are needed to fully understand its mechanisms of action and potential therapeutic benefits.

## 1. Introduction

Hashimoto’s thyroiditis (HT), also known as chronic lymphocytic thyroiditis and Hashimoto’s disease, is an autoimmune thyroid disease (AITD) with a complex etiopathology, which includes genetic (e.g., major histocompatibility complex (MHC) genes) and environmental factors, such as past infections, medications, and smoking, and the level of microelements, such as iodine, iron, and selenium [1,2]. With an incidence of 0.3–1.5 cases per 1000 people, HT is the most common cause of hypothyroidism in iodine-replete areas [3].

Despite much research on its immunopathology, HT remains an incurable disease with an unpredictable nature, often leading to lymphocytic destruction of the thyroid gland and the need for thyroid hormone replacement for life [4,5]. However, some studies showed that interventions within modifiable risk factors might improve immunological and clinical outcomes in HT patients. This includes, among other things, dietary changes, stress management, selenium supplementation, and vitamin D supplementation [6,7,8,9].

In this article, we aim to review the mechanisms of the immunomodulatory activity of vitamin D and its impact on the autoimmune process in Hashimoto’s thyroiditis.

## 2. Material and Methods

The systematic literature research used Pubmed, SCOPUS, and Web of Science databases. The following keywords were used, in combinations or individually: vitamin D, thyroiditis, Hashimoto, autoimmune thyroid disease, immune system, cytokines, and supplementation. The database search focused on original research and meta-analyses presenting studies conducted on humans and animals, published between January 1999 and March 2023. For the trials which would present the association between vitamin D concentration and immunological parameters in HT in humans, as well as changes in immune markers after vitamin D supplementation, we researched the last decade (2013–2023), excluding case reports, narrative reviews, editorials, and commentaries.

## 3. Vitamin D

### 3.1. Sources, Metabolism, and Function

Vitamin D is a fat-soluble vitamin that plays a crucial role in bone health and calcium regulation in the body. There are two main forms of vitamin D: vitamin D2 (ergocalciferol) and vitamin D3 (cholecalciferol). Vitamin D2 is found in some plant foods, while vitamin D3 is synthesized in the skin upon exposure to sunlight [10]. When vitamin D is consumed or synthesized, it first undergoes hydroxylation in the liver, where it is converted into 25-hydroxyvitamin D (25(OH)D), which is commonly used as an indicator of vitamin D status in the body. Next, 25(OH)D is transported to the kidneys, where it undergoes a second hydroxylation step to become the biologically active form of vitamin D known as calcitriol, also known as 1,25-dihydroxyvitamin D (1,25(OH)2D) [11].

Calcitriol then binds to vitamin D receptors (VDRs) in various tissues throughout the body, including the bones, intestines, and kidneys, to regulate calcium and phosphate metabolism. In the intestines, calcitriol increases the absorption of calcium and phosphorus, while in the kidneys, it increases calcium reabsorption and phosphate excretion. Calcitriol helps regulate bone remodeling and mineralization, ensuring proper bone growth and maintenance [12].

In addition to its role in bone health, research has proved its pleiotropic effects which are significant for disease prevention [13]. Many studies have linked vitamin D to various health benefits, including mood regulation, reduced risk of chronic cardiovascular diseases, immune function, and others [14,15,16,17]. Proper levels of vitamin D are essential in the prevention of musculoskeletal [18], cardiovascular [19], dementia [20,21], cancer [22], autoimmune [15], metabolic [23], PCOS [22,24], or even kidney diseases [25]. Examples of pleiotropic actions with their clinical significance in different diseases are shown in Table 1.

The pleiotropic actions of vitamin D are mainly considered to come from acting on the genomic level, directly regulating gene expression via VDR/RXR (vitamin D receptor/retinoid X receptor) complex. However, the research shows it also acts via non-genomic, rapid pathways, altering gene expression through epigenetic mechanisms. For example, in the immune system, as described by Hii et al. [26], vitamin D can control VDR binding to target proteins like STAT1 (signal transducer and activator of transcription 1) and IKK-β (inhibitor of nuclear factor kappa-B kinase subunit beta). This function enables vitamin D to cross-modulate gene expression mediated by non-vitamin D ligands, such as TNF-α and IFN-γ. Vitamin D may exert control over immunological and antiviral responses through this mechanism. Other non-genomic mechanisms of vitamin D action, concerning, among other things, interaction with PDIA3 (protein disulfide isomerase family member 3) and rapid intracellular calcium regulation via L-VGCC (L-type voltage-gated calcium channel) are speculated to play a role in proper bone, muscle, and brain development [27,28].

While sunlight is the most efficient way to obtain vitamin D, it can also be found in certain foods such as fatty fish, egg yolks, dairy, and grain products [29]. However, obtaining enough vitamin D through diet or sunlight alone can be challenging for numerous high-risk groups, which include, among other things, individuals over 65 years of age, dark-skinned populations, patients with cancer, autoimmune diseases, malabsorption syndromes, cardiovascular diseases, diabetes, and people with body mass index (BMI) > 30 [30,31]. According to a recent pooled analysis of 7.9 million participants by Cui et al. [32], 15.7% of the global population is vitamin D deficient. At the same time, other studies indicate that the prevalence of vitamin D deficiency in Europe may be as high as 40%, and the prevalence of severe vitamin D deficiency—13% [33]. Because of the scale of the problem, the need for widespread correction of vitamin D status, mainly with supplementation, is often considered a public health problem [34].

### 3.2. The Role of Vitamin D in the Immune System

Vitamin D significantly impacts the immune system, as VDRs are present in many peripheral blood mononuclear cells, including T and B cells and antigen-presenting cells (APCs) [35]. Via genomic pathways, calcitriol influences the transcriptional activity of genes involved in immune cell functioning and regulates processes such as cell differentiation, the cell cycle, programmed cell death, stress response, and fighting infections. For example, Vitamin D may induce the expression of antibiotic peptides, such as CAMP/LL-37 (cathelicidin antimicrobial peptide LL-37), which destroy the cell membranes of bacteria and viruses [36]. Vitamin D also enhances autophagy in macrophages, which helps clear viruses from the cells, including SARS-CoV-2 [37].

Furthermore, vitamin D plays a significant role in preventing autoimmune processes [15,38,39]. Vitamin D downregulates MHC class II and co-stimulatory molecules expressed on dendritic cells (DCs), which are major APCs, thus preventing excessive T-cell activation [40]. Vitamin D also suppresses DC cytokine production and promotes the expression of anti-inflammatory cytokines, such as interleukin 1 (IL-10) [41]. In T cells, vitamin D suppresses the proliferation and differentiation of CD4+ T cells (helper T cells, Th cells) and promotes their differentiation into Th2 cells, which helps maintain Th1/Th2 balance [42]. Vitamin D also inhibits the development of Th17 cells and promotes the differentiation of regulatory T cells (Tregs) that prevent an increased autoimmune response by, among other things, secreting anti-inflammatory cytokines [43]. B cells, which produce antibodies, also express VDR. Vitamin D has been found to affect B cells in various ways, including the inhibition of naive B cell differentiation or maturation to plasma cells, which may potentially reduce autoantibody production [44].

It should be emphasized that the positive influence of vitamin D on the immune system at the cellular and molecular level also translates into improved clinical outcomes for patients with autoimmune diseases. Several clinical trials and observational studies have demonstrated that vitamin D supplementation may benefit the prevalence and disease activity of multiple autoimmune conditions, e.g., rheumatoid arthritis [45], inflammatory bowel disease [46], and vitiligo [47]. For example, the VITAL randomized controlled trial from 2022, which included 25,871 participants from the United States, showed that vitamin D supplementation of 2000 IU/day for five years reduced the risk of developing any autoimmune disease by 22% [48]. Multiple meta-analyses also proved that vitamin D deficiency increases the risk of developing various autoimmune conditions, including AITD [49,50,51,52,53].

## 4. Recent Findings in Hashimoto’s Thyroiditis Immunopathology

### 4.1. Etiological Factors Affecting the Development of HT

Despite many studies leading to a better understanding of autoimmune processes at the cellular and molecular level, the exact causes and triggers that initiate the thyroid destruction process in HT remain unknown. Multiple factors have been proven to impact the risk of HT, including exposure to environmental components (e.g., infections, irradiation, selenium), the presence of different diseases (e.g., diabetes, allergic rhinitis), and differentially expressed genes [54,55,56]. Genetic predisposition includes MHC genes (e.g., HLA-DR3) and genes regulating immune responses like IL-7R (Interleukin 7 receptor) gene, OR2J3 (olfactory receptor 2J3) gene, or CTLA4 (cytotoxic T-lymphocyte-associated protein 4) gene. This proves the complex etiology of the disease, for which there is still no effective causal treatment.

We know that the core of the autoimmune process in HT is a breakdown in self-tolerance to thyroid autoantigens, which leads to thyroid destruction by infiltration of CD4+ T cells, macrophages, and plasma cells, which produce auto-antibodies against thyroid peroxidase (anti-TPO) and thyroglobulin (anti-Tg) [57]. Therefore, the detection of elevated titers of these antibodies is generally used for HT diagnosis [58].

### 4.2. Immunopathological Processes in HT on the Level of Cells and Cytokines

From the morphopathological perspective, HT disease typically leads to thyroid enlargement with nodule development. In HT, there is an extensive infiltration of the parenchyma by a mononuclear inflammatory infiltrate containing small lymphocytes, plasma cells, and well-developed germinal centers. The CD4:CD8 ratio in the infiltrate is 4:1. The atrophy of the colloid bodies is lined by Hürthle cells [59,60].

Many studies show that, from the immunological viewpoint, the main contributors to the development of HT are (1) Th1/Th2 cell imbalance and (2) Th1 cell activity enhancement [16], which lead to disturbances in the complex interplay between different immune components. The characteristics of thyroid autoimmunity in HT on the cellular and molecular levels were reviewed by Luty et al. in detail [61].

First, the presence of environmental/genetic factors leads to the activation of APCs, mainly DCs, which present allo- and autoantigens to naive CD4+ T cells, leading to their differentiation into Th1, Th2, Th17, Th22, or Tregs (Figure 1). Second, cytokines produced by Th1, including IL-12 and IFN-γ, induce the expression of MHC II on thyroid cells, further promoting the differentiation of the naive CD4+ T cells into Th1. Finally, Th1 cells, through IFN-γ, IL-2, and TGF-β, induce the activation of CD8+ T cells (cytotoxic T cells, Tc cells).

CD8+ T cells, through secreted perforins and granzymes or via Fas-FasL cascade [62], destroy thyroid cells, which leads to the release of proinflammatory cytokines and chemokines. This leads to an amplification feedback loop that initiates and sustains the immune process. Cytokines released by thyrocytes contribute to (1) the migration and activation of pathological Th17 cells (IL-6, TNF-α, IL-1β, and TGF-β) and (2) the suppression of Tregs (IL-6, IFN-γ, IL-8, IL-1β, and CXCL10), which usually inhibit excessive T-cell-mediated cytotoxicity [63]. Moreover, the microenvironment in the infiltrated thyroid promotes the differentiation of proinflammatory Th22 cells. The destruction of the thyroid gland and the strengthening of the autoimmune process also occur through the humoral response. Infiltrating B cells (triggered by Th2 cytokines, i.e., IL-4, IL-5, and IL-10) release autoantibodies (mainly anti-TPO and anti-Tg), which can further lead to the destruction of thyrocytes in the mechanism of antibody-dependent cell-mediated cytotoxicity (ADCC) [61].

## 5. Role of Vitamin D in the Immunopathology of Hashimoto’s Thyroiditis

### 5.1. Immunomodulatory Potential of Vitamin D in HT

Rui et al. [16] presented four potential mechanisms in which Vitamin D can contribute to the inhibition of the immune process in HT: (1) prevention of DC-dependent T-cell activation; (2) down-regulation of HLA class II gene expression in the thyroid; (3) influence on B-cells and (4) restoration of the Th17/Tregs ratio.

Most subsets of dendritic cells (e.g., conventional DCs and Langerhans cells) have vitamin D receptors on their surface [64]. It has been proven that the 1,25(OH)2D3-VDR complex can inhibit the expression of various proinflammatory cytokines from DCs that activate T cells (IL-2, IL-5, IL-17, IL-12, IL-23, IL-6, TNFα, INF-γ, CCL5, and CCL17) while enhancing the expression of IL-10 and IL-8. These effects may potentially contribute to the improvement of HT by (1) a shift from the Th1 profile towards Th2 (2), a decrease in pathological Th17 responses (3), and a decrease in a cytokine-mediated immune response [64,65].

In contrast to the healthy population, the follicular thyroid cells of HT patients may express MHC class II molecules, which are crucial for presenting antigens to CD4+ T cells. In that manner, thyroid cells can act as APCs by presenting autoantigens to T cells and activating them. Induction of MHC class II on follicular cells by, i.e., IFN-γ, and IL-12, can contribute to the autoimmune process in HT [61]. 1,25(OH)D can reduce MHC II expression, thus preventing T-cell activation and proinflammatory cytokine response [15].

B cells play a role in the pathogenesis of HT mainly by producing antibodies, anti-TPO, and anti-Tg, which are thyroid self-antigens. Antibodies contribute to the apoptosis of thyroid follicular cells in the mechanism of antibody-dependent cell-mediated cytotoxicity [66]. Although the particular role of vitamin D on ADCC in HT is not fully understood, it has been suggested that insufficient 25(OH)D concentration are associated with increased B-cell proliferation, differentiation, and antibody titers in autoimmune diseases, such as SLE and MS [67,68]. Vitamin D has been proven to exert inhibitory effects on plasma cell generation, which can, in turn, contribute to decreased immunoglobulin production [69]. Some studies demonstrated that 1,25D-treated Th cells may suppress B-cell differentiation, proliferation, and antibody production [40]. Many studies proved the association between low vitamin D status and antithyroid antibody titers and a decrease in antithyroid antibody titers after treating vitamin D deficiency. These effects have been confirmed in meta-analyses [53,70,71,72].

Although Th17 and Tregs differentiate from a common precursor, the naive CD4+ T cells, have different, even opposite, functions. Th17 cells express mainly proinflammatory activity, which contributes to the development of inflammation and autoimmune disorders, whereas Tregs modulate the immune system and maintain tolerance to self-antigens which prevents autoimmunity [73].

Many studies confirmed the critical role of increased Th17/Treg ratio in the pathogenesis of various autoimmune disorders, including AITD [74,75]. Vitamin D inhibits the differentiation of naive T cells into Th17 while increasing the levels of Tregs, which restores the Th17/Treg ratio in the body [76,77]. Moreover, vitamin D can inhibit the secretion of cytokines (mainly IL-17) from Th17 cells [78], potentially reducing the Th17 cell-induced thyroid inflammation.

### 5.2. Association between Vitamin D, the Occurrence of HT, and Antibody Levels

The association between vitamin D and HT remains controversial. Many studies to date investigated this topic in various populations, and the results still have some inconsistency. Some work, including observations from the Croatian Biobank of HT patients and other comparative studies, did not detect an association between vitamin D levels and the prevalence of HT [79,80,81]. However, large-scale studies, including systematic reviews and meta-analyses, confirm the association between low vitamin D and HT. Meta-analysis of observational studies by Taheriniya et al. [53] showed lower vitamin D levels in patients with HT than in healthy subjects. Similar results were shown in meta-analyses by Wang et al. [71] and Štefanić and Tokić [72]; HT patients were more likely to have lower 25(OH)D than a healthy population. In studies by Kim et al. [82,83] on the Korean population, including a nationwide survey of 4181 participants, vitamin D deficiency was significantly associated with a high prevalence of thyroid autoimmunity. Other studies showed that low 25(OH)D correlates with HT in children and adolescents [84], as well as in obese subjects [85]. Moreover, some studies suggest an association between serum 25(OH)D levels and the clinical presentation of HT, including the severity of hypothyroidism and the prevalence of mild cognitive impairment [79,86].

Apart from the correlation between vitamin D status and the prevalence of HT, several studies investigated the association between vitamin D and antithyroid antibody levels. In the comparative study by Aktaş et al. [87], there was a negative correlation between 25(OH)D level and anti-TPO in 130 patients diagnosed with HT. Similar results were obtained by Bozkurt et al. [88], who also demonstrated that vitamin D deficiency severity correlated with the duration of HT, thyroid volume, and antibody levels. Interestingly, Sayki Arslan et al. [89] also discovered that anti-TPO positivity was significantly more common in healthy subjects (HT not diagnosed) with vitamin D deficiency compared to those with a normal 25(OH)D level.

### 5.3. Association between Vitamin D Levels and Immunological Parameters in HT

Most studies that evaluated the immunity of HT concerning vitamin D focus on thyroid autoantibody titers as markers of the autoimmunological process. However, the potential impact of vitamin D may concern various other immunological mechanisms. Few studies assessed the association of vitamin D and HT in the context of different immune pathways, with varying results. For example, in HT patients, Botehlo et al. [90] showed no significant correlation between 25(OH)D status and IL-2, IL-4, and IFN-γ serum levels. However, a positive correlation was observed between vitamin D and IL-17, TNF-α, and IL-5. According to the authors, the lower TNF-α and IL-17 levels, which correlated with low vitamin D status, could be explained by the control of cytotoxicity by long-time treatment of HT. As stated by Korzeniowska et al. [91], levothyroxine may contribute to stabilizing the inflammatory process in HT. In a study by Wencai Ke et al. [92], serum 25(OH)D levels were not associated with IL-4, IL-17, and TNF-α in newly diagnosed or treated patients with HT. However, vitamin D concentrations were relatively deficient in those subjects.

Feng et al. [93] discovered that, amongst Chinese children with HT, serum IL-21 concentration was positively correlated with antithyroid antibodies, while the serum concentration of 25(OH)D had a significant negative correlation with serum IL-21. According to the investigators, the results showed that vitamin D levels and IL-21 might be involved in the occurrence and development of HT.

It is worth noting that the results by Hisbiyah et al. [94], who investigated children with Down syndrome and HT. Contrary to most studies, they found a positive correlation between vitamin D and antithyroid antibody levels. They also found no correlation between vitamin D and NF-κB (nuclear factor-kappa B), suggesting that vitamin D could not affect NF-κB, a transcription factor whose pathway, according to Giuliani et al. [95], can also play a role in thyroid autoimmunity. However, the authors concluded that vitamin D could suppress IFN-γ, which is involved in, i.e., the suppression of Tregs expression and the activation of CD8+ T cells in HT. [61]

Roehlen et al. [96] presented even more insights into the mechanisms of vitamin D immunoregulation. They investigated if the immunoregulatory function of vitamin D can be related to FOXO3a gene polymorphisms and SIRT1 (sirtuin 1 histone deacetylase) in HT and DTC (differentiated thyroid cancer). SIRT1 and FOXO3a are proteins that regulate cellular processes related to aging and disease [97]. FOXO3a is a transcription factor that regulates the expression of multiple genes associated with, i.e., cell proliferation, apoptosis, and cellular stress [98], while SIRT1 is a deacetylase that modifies the activity of various proteins, e.g., p53, NF-κB, and FOXO3a [99]. SIRT1 activates FOXO3a through deacetylation, leading to the upregulation of genes involved in stress resistance and longevity. In vitro, vitamin D exerted an anti-proliferative effect in Th cells that was blocked by SIRT1 inhibition and accompanied by elevated FOXO3a gene expression. The authors concluded that the SIRT1-FOXO3a axis is one of the downstream targets of vitamin D immunoregulatory effects. Moreover, they identified two single nucleotide polymorphisms in the FOXO3a gene (rs9400239T and rs4945816C) that may constitute genetic risk factors for HT [96].

Tokic et al. [100] discovered that T cells from HT patients exhibited lower CTLA4, CD28, and CD45RAB gene expression than healthy controls. All these molecules may play a role in thyroid autoimmunity. The CD28/T-cell receptor (TCR)/CTLA4 complex regulates T-cell homeostasis and tolerance in HT, while CD45 protein tyrosine phosphatase (PTPase) cooperates with the vitamin D receptor to interact with the TCR complex and influence the Th1/Th17/Treg pathways that are critical to the development of HT.

The association between vitamin D status, antithyroid antibody levels, and different immunological parameters in different studies is summarized in Table 2.

### 5.4. Changes in Immunological Parameters and HT Outcomes after Vitamin D Supplementation

Several studies investigated immunological and inflammatory changes after vitamin D supplementation in patients with HT. Most results show a significant improvement in markers of immunity after the intervention. Krysiak et al. [101,102,103] conducted experiments amongst various HT populations in Poland (e.g., men with HT and alopecia [101], men with HT and testosterone deficiency [102], and euthyroid women with HT [102,103]), in which they assessed anti-TPO and anti-Tg levels after six months of 2000–4000 IU daily vitamin D supplementation. In these studies, 25(OH)D concentration increased (often achieving levels exceeding 30 ng/mL, which are adequate, according to Polish, as well as Central and European guidelines [104,105], and antithyroid antibody levels significantly decreased. The reduction in antibody titers was also present in the subjects with normal baseline vitamin D status [7]. In addition, researchers discovered many factors that impacted the effect of vitamin D on antithyroid antibodies. Dehydroepiandrosterone, simvastatin, and selenomethionine potentiated the effect, while hyperprolactinemia and a gluten-free diet (GFD) attenuated it [103,106,107,108,109]. GFD inhibiting the reduction in antithyroid antibodies by vitamin D is particularly interesting, as it adds another argument to the discussion about whether a gluten-free diet should be considered for non-celiac autoimmune disorders, which remains a controversial topic [110,111]. Authors explained that GFD patients might have ingested smaller amounts of unsaturated fatty acids, iron, and calcium, contributing to proper intestinal vitamin D absorption [109].

Other studies in different populations also obtained a reduction in antithyroid antibody levels after vitamin D supplementation. For example, Mazokopakis et al. [112] achieved a significant reduction in anti-TPO and anti-Tg with doses from 1200 to 4000 IU every day for 4 months (aiming to maintain serum 25(OH)D levels of 40 ng/mL) in Greek patients. Similar results were also obtained in Indian patients (60,000 IU weekly for 8 weeks reduced anti-TPO by 46.73%) [113] and amongst Iranian children with HT [114]. Noteworthy are also the results from Sinsek et al. [115], who demonstrated a significant reduction in anti-TPO and anti-Tg amongst Turkish HT and GD patients with only one month of 1000 IU daily vitamin D. A meta-analysis by Zhang et al. [70] also showed that vitamin D supplementation reduces anti-TPO and anti-Tg titers, especially with a supplementation duration of over 3 months.

Particularly intriguing are the outcomes of the community-based program conducted in Canada. The program database included 11,017 participants that were provided with vitamin D to achieve 25(OH)D concentrations exceeding 100 nmol/L (>40 ng/mL). After 12 months of follow-up, a significant reduction in antithyroid antibodies was observed [116]. Among subjects with elevated antithyroid antibody levels at baseline and follow-up, 77.5% were within the reference range for anti-TG and 42.2% for anti-TPO. In addition, serum 25(OH)D concentrations ≥ 125 nmol/L were associated with a 32% reduced risk of elevated antithyroid antibodies. Moreover, achieving correct vitamin D concentration was associated with substantial improvement in thyroid function, including reduced TSH levels and the severity of symptoms. This effect was especially amongst subjects with subclinical hypothyroidism, which was reduced by 72% at follow-up.

Although most studies on antithyroid antibodies after vitamin D gave consistent outcomes, it should be mentioned that some investigations showed contrary results. Vondra et al. [117] detected no decrease in antithyroid antibodies after three months of 4300 IU/day 25(OH)D supplementation, which could be explained by their relatively low initial levels. However, in a study by Behera et al. [118], amongst 23 HT patients from a coastal province of India, a significant increase in anti-TPO occurred after weekly 60,000 IU vitamin D supplementation for 8 weeks. The authors stated that they were unable to explain this effect.

Besides antibody levels, several researchers explored how vitamin D supplementation can affect other immunological markers. Robat-Jazi et al. [119] supplemented 40 HT female patients with 50,000 IU of 25(OH)D weekly for 3 months. At the baseline and follow-up, they assessed the serum levels of inflammatory factors, such as TNF-α, IFN-γ, and the chemokine CXCL10 (or IP10, interferon gamma-induced protein 10). IP10 is involved in the activation of T cells and the regulation of immune cell migration and proliferation. It may also contribute to the breakdown of immune tolerance by promoting the activation of autoreactive T cells in the thyroid [61,120]. IP10 is elevated in several other autoimmune diseases, including RA, MS, and SLE [121]. In this study, the serum levels of IFN-γ, TNF-α, and IP10 decreased significantly after the intervention suggesting the potential of vitamin D to control the inflammatory axis of IFNγ-IP10 and TNF-α in HT. A clinical trial conducted by Nodehi et al. [122] assessed the frequency of CD4+ T-cell subsets before and after the 3-month supplementation of 50,000 IU weekly vitamin D. The supplementation provided beneficial immunological effects by causing a significant decrease in Th17/Tr1 (regulatory T cells type 1) ratio. However, the activity of IL-10 in Tr1 cells increased in the vitamin D group, compared to that in the placebo group, with no statistical significance.

The changes in antithyroid antibody levels and different immunological parameters after vitamin D supplementation in different studies are summarized in Table 3.

## 6. Conclusions

In conclusion, the relationship between vitamin D and Hashimoto’s thyroiditis has been the subject of numerous studies with varying results. Recent findings indicate that vitamin D may have an immunomodulatory effect on HT by, among other things, influencing DC-dependent T-cell activation, downregulating HLA class II gene expression in the thyroid, preventing excessive B-cell response, and balancing the Th17/Treg cell ratio. While laboratory insights into vitamin D function in HT are valuable, it is also crucial to examine the real-world clinical outcomes, including associations between vitamin D and symptoms, thyroid, and immune system function, as well as the impact of vitamin D on disease manifestation and progression. While some research has shown no association between 25(OH)D concentration and the prevalence of HT, several large-scale studies, systematic reviews, and meta-analyses have confirmed a link between low 25(OH)D levels and HT. Additionally, numerous studies have demonstrated a negative correlation between vitamin D levels and antithyroid antibody levels.

Besides the correlation between vitamin D status and HT prevalence, research has explored the association between vitamin D and other immunological parameters in HT, such as cytokines and T-cell subsets. While the results have been mixed, some studies have shown a significant negative correlation between 25(OH)D concentration and inflammatory cytokines.

Research on the effects of vitamin D supplementation in HT patients has shown promising results, with most studies reporting a significant improvement in immunological markers after the intervention. However, a few studies have yielded contrasting findings. Overall, the available evidence suggests that vitamin D supplementation may have beneficial immunomodulatory effects in HT patients, but further research is necessary to determine the optimal dosing, duration, and potential interactions with other treatments or dietary factors.

Future studies need to continue investigating the complex relationship between vitamin D and HT and the impact of vitamin D supplementation on various immunological markers and clinical outcomes. This knowledge will contribute to a better understanding of the role of vitamin D in HT pathogenesis and inform potential therapeutic strategies for individuals with HT.

## Figures and Tables

**Figure 1 nutrients-15-03174-f001:**
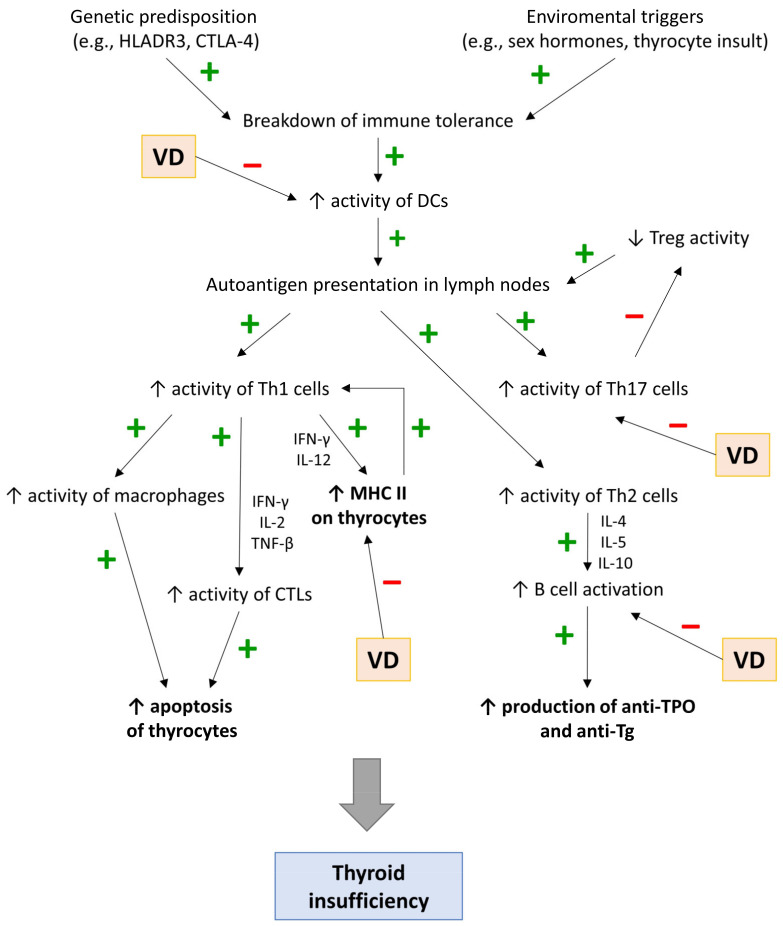
Immunopathomechanisms of Hashimoto’s thyroiditis.

**Table 1 nutrients-15-03174-t001:** Examples of possible pleiotropic actions of vitamin D and their potential clinical significance.

Disease Group	Vitamin D Functions	Disease Prevention
Musculoskeletal diseases [18]	- Promotion of osteoblast proliferation; - DMP1 and BSP synthesis regulation; - Activation of muscle protein synthesis.	- Rickets, osteoporosis, and fractures; - Sarcopenia.
Cardiovascular diseases [19]	- Regulation of calcium metabolism, including intracellular calcium concentration; - RAAS regulation.	- Cardiomyocyte dysfunction; - Hypertension.
Brain diseases [20,21]	- Regulation of dopaminergic development; - Reduction in amyloid-induced cytotoxicity.	- Schizophrenia; - Alzheimer’s disease, dementia.
Cancers [22]	- VEGF inhibition; - Blocking cell cycle at G0/G1 stage.	- Prostate cancer; - Melanoma; - Head and neck cancers.
Immune-mediated diseases [15]	- Preventing excessive T-cell activation; - Inhibits the development of Th17 cells; - Promotes the differentiation of regulatory T-cells.	- RA; - AITD; - SLE; - MS.
Metabolic diseases [23]	- Reduction in insulin resistance; - Regulation of adipogenesis.	- Diabetes mellitus; - Obesity.
Female reproductive system diseases [22,24]	- Progesterone production stimulation; - Reduction in insulin resistance.	PCOS
Renal system diseases [25]	- Prevention of renal fibrosis, apoptosis, and inflammation.	CKD

AITD, autoimmune thyroid disease; BSP, bone sialoprotein; CKD, chronic kidney disease; DMP, dentin matrix acidic phosphoprotein; MS, multiple sclerosis; PCOS, polycystic ovaries syndrome; RA, rheumatoid arthritis; RAAS, renin–angiotensin–aldosterone system; SLE, systemic lupus erythematosus; VEGF, vascular endothelial growth factor.

**Table 2 nutrients-15-03174-t002:** The association between vitamin D status, antithyroid antibody levels, and different immunological parameters in different studies.

Author, Year	Type of Study	Number of Participants	Main Findings
Bozkurt et al., 2013 [88]	Case–control study	- 180 treated HT patients; - 180 newly diagnosed HT patients; - 180 HC.	- 25(OH)D levels of HT patients were significantly lower than controls; - 25(OH)D deficiency severity correlated with duration of HT, thyroid volume, and antithyroid antibody levels.
Evliyaoğlu et al., 2015 [84]	Case–control study	- 90 HT patients; - 79 HC.	The prevalence of 25(OH)D deficiency in HT patients was significantly higher than that in the control group
Arslan et al., 2015 [89]	Cross-sectional study	155 HC	- Anti-TPO positivity was significantly more common in those with 25(OH)D deficiency, as compared to those with a normal 25(OH)D level. - A significant weak inverse correlation between anti-TPO positivity and the 25(OH)D level.
Wang et al., 2015 [71]	Meta-analysis	(a) The continuous 25(OH)D by AITD * status: - 1782 AITD patients; - 1821 HC. (b) The presence of 25(OH)D deficiency: - 994 AITD patients; - 1035 HC.	HT patients had lower 25(OH)D levels and were more likely to have a 25(OH)D deficiency
Kim et al., 2016 [82]	Retrospective cross-sectional study	776 HT patients with measured 25(OH)D	25(OH)D insufficiency was associated with HT, especially overt hypothyroidism
Kim et al., 2017 [83]	Cross-sectional study; a nationwide survey	4181 participants	Anti-TPO positivity was more prevalent in the 25(OH)D deficient group than in insufficient and sufficient 25(OH)D groups
Wencai Ke et al., 2017 [92]	Cross-sectional study	- 124 HT patients; - 51 GD patients.	25(OH)D levels were not associated with thyroid function, antithyroid antibodies, and serum cytokines IL-4, IL-17, and TNF-α in patients with AITD *
Tokic et al., 2017 [100]	Cross-sectional study	- 45 HT patients; - 13 HC.	Nominally higher expression levels of VDR mRNA were found in T cells of healthy controls when compared to the HT patients
De Pergola et al., 2018 [85]	Cross-sectional study	261 overweight and obese subjects	25(OH)D deficiency is significantly associated with HT in overweight and obese subjects
Botelho et al., 2018 [90]	Cross-sectional study	- 88 HT patients; - 71 HC.	A positive correlation between 25(OH)D and fT4, IL-17, TNF-α and IL-5 in HT group
Jun Xu et al., 2018 [86]	Case–control study	- 194 patients with HT; - 200 HC.	- HT patients with MCI had significantly lower 25(OH)D levels than patients without MCI. - 25(OH)D levels of ≤34.0 and ≥47.1 nmol/L were significantly associated with cognitive impairment in patients with HT.
Aktaş, 2019 [87]	Retrospective cohort study	130 HT patients	- 25(OH)D deficiency was associated with HT. - A negative correlation between 25(OH)D levels and anti-TPO.
Feng et al., 2020 [93]	Cross-sectional study	- 36 HT patients; - 54 GD patients; - 30 HC.	- 25(OH)D was lower, while IL-21 was higher in HT patients than in control. - 25(OH)D was negatively correlated with anti-TPO and anti-Tg. - IL-21 concentration was positively correlated with anti-TPOAb, anti-Tg, and TRAb in the HT group. - 25(OH)D had a significant negative correlation with serum IL-21 concentration in HT.
Štefanić and Tokić, 2020 [55]	Meta-analysis	- 2695 HT patients; - 2263 HC.	Lower serum 25(OH) in HT compared to healthy controls
Cvek et al., 2021 [79]	Case–control study; observations from biobank data	- 461 HT patients; - 176 HC.	- No significant differences in 25(OH)D levels between HT patients and controls. - A subtle decrease in 25(OH)D levels associated with overt HT, compared to mild HT.
Hanna et al., 2021 [80]	Case–control study	- 112 HT patients; - 48 hypothyroid non-HT controls.	25(OH)D level was statistically indifferent between HT and control groups
Taheriniya et al., 2021 [53]	Meta-analysis	- 1375 HT patients; - 1065 HC.	Significantly lower 25(OH)D level among HT patients compared to healthy controls
Hisbiyah et al., 2022 [94]	Cross-sectional study	80 Down syndrome patients (children)	- Participants with sufficient 25(OH)D had significantly higher anti-TPO and anti-Tg - 25(OH)D levels were significantly negatively correlated with IFN-γ and positively correlated with anti-TPO-Ab and anti-Tg.
Filipova et al., 2023 [81]	Prospective case–control study	- 57 HT patients; - 41 HC.	No significant association between 25(OH)D and thyroid autoantibodies, thyroid hormones, and thyroid volume

25(OH)D, 25-hydroxycholecalciferol; AITD, autoimmune thyroid disease; anti-TPO, antithyroid peroxidase antibodies; fT4, free thyroxine; GD, Graves’ disease; HC, healthy controls; HT, Hashimoto’s thyroiditis; MCI, mild cognitive impairment; TRAb, TSH receptor antibody; VD, vitamin D; * AITD patients included Graves’ disease and Hashimoto’s thyroiditis patients.

**Table 3 nutrients-15-03174-t003:** Changes in antithyroid antibody levels and different immunological parameters after vitamin D supplementation in different studies.

Author, Year	Dose and Supplementation Duration	Changes in 25(OH)D Levels	Other Changes in Immunological Parameters	Changes in Anti-TPO Titers	Changes in Anti-Tg Titers
Mazokopakis et al., 2015 [112]	1200–4000 IU daily, aiming to achieve 25(OH)D concentration of >40 ng/mL, 4 months	⬆		⬇	⬇
Chaudhary et al., 2016 [113]	60,000 IU weekly, 8 weeks	⬆		⬇	
Simsek et al., 2016 [115]	1000 IU daily, 1 month	⬆		⬇	⬇
Mirhosseini et al., 2017 [116]	Doses modified with the aim to achieve 25(OH)D concentration of >40 ng/mL, 1 year	⬆	CRP ⬇	⬇	⬇
Vondra et al., 2017 [117]	4300 IU daily, 3 months	⬆	CRP ⬌	⬆	⬆
Nodehi et al., 2019 [122]	50,000 IU weekly, 3 months	⬆	- Th17/Tr1 ratio ⬇ - IL-10 ⬆		
Aghili et al., 2020 [114]	Varied depending on initial and rechecked 25(OH)D concentrations	⬆		⬇	⬇
Behera et al., 2020 [118]	60,000 IU weekly, 2 months, then 60,000 IU monthly, 4 months	⬆		⬆	
Krysiak et al., 2016–2022 [7,101,102,103,106,107,108]	2000–4000 IU daily for 6 months	⬆		⬇	⬇
Krysiak et al., 2022 [109]	4000 IU daily for 6 months	⬆	CRP ⬌	⬇	⬇
Robat-Jazi et al., 2022 [119]	50,000 IU weekly, 3 months	⬆	- IFN-γ ⬇ - IP10 ⬇ - TNF-α ⬇	⬇	

25(OH)D, 25-hydroxycholecalciferol; CRP, C-reactive protein; IL-10; interleukin 10; IFN-γ, interferon gamma; IP10, interferon gamma-induced protein 10; Th17, T helper 17 cell; TNF-α, tumor necrosis factor alpha; Tr1, type 1 regulatory T cell; ⬆, an increase; ⬇, a decrease; ⬌, no significant change.

## Data Availability

Not applicable.

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
