# Peer review of "Impact of Vitamin D on Immunopathology of Hashimoto’s Thyroiditis: From Theory to Practice"

_nutrients, 2023, doi:10.3390/nu15143174_

Round 1
Reviewer 1 Report
Dear Authors,
I can say that it was not so much complicated to read your work, what means that it was well written - congrats. As the reviewer, I want to say what you should consieder to be changed.
1) words as "level" or "levels" or "vitamin D level" are misleading since you did not provided evidence related to cholecalciferol but on "25(OH)D concentration". The last word/phrase should be used in the whole text of your paper.
2) You are from Poland, Poland is well known of "thyroid" studies, and had also guidelines papers not only for Central Europe but also for Poland.
3) The pleiotropic action, at least some risk-groups.... should be emphesized more in your paper.
good english
Author Response
Answers to the first Reviewer. Corrections in the text are marked in green.
"Dear Authors,
I can say that it was not so much complicated to read your work, what means that it was well written - congrats. As the reviewer, I want to say what you should consieder to be changed."
Thank you for comments and suggestions.
"Words as "level" or "levels" or "vitamin D level" are misleading since you did not provided evidence related to cholecalciferol but on "25(OH)D concentration". The last word/phrase should be used in the whole text of your paper."
Thank you for that clarification. After the correction, "25(OH)D concentration" is used instead of "vitamin D level(s)" wherever the 25(OH)D measurement is concerned. However, in some sentences, we left "vitamin D status" unchanged, as we explained in section 2. that "25-hydroxyvitamin D (25(OH)D) […] is commonly used as an indicator of vitamin D status in the body".
"You are from Poland, Poland is well known of "thyroid" studies, and had also guidelines papers not only for Central Europe but also for Poland."
Of course. We added the citation for Polish updated Vitamin D Guidelines for complementation (reference no. 108, page 20).
"The pleiotropic action, at least some risk-groups.... should be emphesized more in your paper."
To introduce the multidirectional effects of vitamin D, we summarized the most important pleiotropic actions in Table 1. We also gave examples of high-risk groups for vitamin D deficiency, as well as we underlined the worldwide scale of the problem (page 2).
Reviewer 2 Report
General thoughts
1. An undoubted advantage of the work is the rich and up-to-date literature (65% of items come from the years 2018 - 2023). The work is properly planned, although some minor changes are suggested to the Authors, such as merging subsection 4.2 with subsection 4.1 and moving the fragment between lines 191-206 to the end of the current subsection 4.3. It is also suggested to supplement the subtitle in p. 4.4: "Studies on changes (what changes?) after vitamin D supplementation".
2. It is better, if one name of the described disease is used throughout the text. In the content of individual points, the use of indicated substitutes or abbreviations, such as Hashimoto thyroiditis or HT is acceptable. However, new terms, especially terms not mentioned before should not be used, especially in the titles or subtitles of the work, as in the title p. 3: "Hashimoto".
3. The introduction should mention the criteria according to which the authors selected the literature constituting the basis for the meta-analysis.
Detailed remarks
2.1
Vitamin D exerts its activity via genomic or nongenomic pathway. The first one, via VDR and transcriptional complex RXR was described in the work. However the other one has been unmentioned. It is interesting that this pathway is activated with noted mediation of, among others, thyroid hormones. The authors were probably guided in the description of the mechanisms of action by the relationship with the subject of the article, but the current version of the subsection suggests that the VDR route for calcitriol is the only one and it may be worth mentioning all possible signaling pathways activated by calcitriol.
3.2 Figure 1 (line 143) should again be referenced at the end of section 3.2.
4.4 Line 330 - the abbreviations SLE and MS should be explained.
Table 1 and Table 2
Subsequent items of Table 1 should be arranged chronologically.
Table 2
Subsequent items of Table 2 should be arranged chronologically. The content of the Table should be covered in the point that Table refers to (4.4 in this case). Meanwhile, in this section, the content of the point is not a description of the Table, but concerns studies other than those presented in the form of the Table 2.
Table 2 should include dosing schedules and doses of vitamin D used in individual studies.
Both Tables 1 and 2 should include quantitative data to illustrate the intensity of any effect, and statistical evaluation of the results (at least p-value). Without these data, which greatly facilitate the assessment of the results of supplementation or vitamin D status, the key information on the subject regarding the influence of vitamin D on HT is not sufficiently emphasized.
Author Response
Answers to the second Reviewer. Corrections in the text are marked in blue.
"An undoubted advantage of the work is the rich and up-to-date literature (65% of items come from the years 2018 - 2023). The work is properly planned, although some minor changes are suggested to the Authors, such as (1) merging subsection 4.2 with subsection 4.1 and (2) moving the fragment between lines 191-206 to the end of the current subsection 4.3. (3)It is also suggested to supplement the subtitle in p. 4.4: "Studies on changes (what changes?) after vitamin D supplementation".
We are grateful for the kind words about our work. Thank you for comments and suggestions. However, we would prefer to keep paragraphs 5.1 and 5.2 (earlier paragraphs 4.1 and 4.2) separated because paragraph 5.1. describes potential (proposed) immunomodulatory activity of VD in HT, while next two paragraphs (5.2 and 5.3) describes associations between VD levels and HT prevalence and autoantibodies (5.2) and immunological parameters, e.g. T cells, cytokines (5.3). For the same reason, we would like to leave mentioned fragment where it is now. Although the mentioned paragraph "The association between vitamin D and HT remains controversial. Many studies to date investigated this topic in various populations, and there is still some inconsistency in the results." may contain summative overtone, we decided to put it at the beginning of the section. We believe that it introduces the reader to the topic of vitamin D and HT in the context of the most often used clinical markers for HT diagnosis for clinicians – anti-thyroid antibodies. Our purpose was to first show the reader the current state of knowledge using these parameters, then, in the next section, show how this impact is studied in more in-depth markers of immune processes. After that, we wanted to summarize the effects of actual treatment with vitamin D.
Finally, we supplemented the title into "Changes in immunological parameters and HT outcomes after vitamin D supplementation" (page 11).
"2. It is better, if one name of the described disease is used throughout the text. In the content of individual points, the use of indicated substitutes or abbreviations, such as Hashimoto thyroiditis or HT is acceptable. However, new terms, especially terms not mentioned before should not be used, especially in the titles or subtitles of the work, as in the title p. 3: "Hashimoto"."
Thank you for noticing that, we corrected the title by adding "thyroiditis" (page 4, paragraph 4).
"The introduction should mention the criteria according to which the authors selected the literature constituting the basis for the meta-analysis."
Thank you for that suggestion. The purpose of this particular work was rather to briefly show the complexity of research of vitamin D influence on HT, introduce the different populations which have been studied and present the variety of ways that this influence can be measured on immunological level. It was not our main intent to try to integrate the data in the meta-analysis. However, we did perform a selection process, about which the information is now supplemented into the article (paragraph 2, Material and methods).
"Vitamin D exerts its activity via genomic or nongenomic pathway. The first one, via VDR and transcriptional complex RXR was described in the work. However the other one has been unmentioned. It is interesting that this pathway is activated with noted mediation of, among others, thyroid hormones. The authors were probably guided in the description of the mechanisms of action by the relationship with the subject of the article, but the current version of the subsection suggests that the VDR route for calcitriol is the only one and it may be worth mentioning all possible signaling pathways activated by calcitriol."
Thank you for that suggestion, we briefly introduced the examples of non-genomic actions with potential clinical significance (page 2, paragraph 3, Vitamin D).
"3.2 Figure 1 (line 143) should again be referenced at the end of section 3.2."
Thank you for noticing. Actually, we forgot to reference Figure 1 in the text. We added the reference in the paragraph 4.2 (earlier paragraph 3.2).
"4.4 Line 330 - the abbreviations SLE and MS should be explained."
We now added these abbreviations under Table 1, where there are used the first time.
"Subsequent items of Table 1 should be arranged chronologically. Subsequent items of Table 2 should be arranged chronologically."
Thank you, we now arranged the subsequent items of Table 2 and 3 (earlier Table 1 and 2) chronologically.
"The content of the Table should be covered in the point that Table refers to (4.4 in this case). Meanwhile, in this section, the content of the point is not a description of the Table, but concerns studies other than those presented in the form of the Table 2."
Thank you for this comment. Indeed, there are discrepancies between the text in both tables. We reviewed the literature in the text and tables and corrected these inaccuracies.
"Table 2 should include dosing schedules and doses of vitamin D used in individual studies."
Thank you for that note, the doses and duration of studies is now included in Table 2 (now Table 3).
"Both Tables 1 and 2 should include quantitative data to illustrate the intensity of any effect, and statistical evaluation of the results (at least p-value). Without these data, which greatly facilitate the assessment of the results of supplementation or vitamin D status, the key information on the subject regarding the influence of vitamin D on HT is not sufficiently emphasized."
Thank you for that note. We greatly appreciate the value that quantitative data adds to the clarity of results analysis. However, we believe that the tables in the present state suit the context of our article.
In Table 1 (now Table 2), we chose a descriptive way to show results instead of quantitative data, mainly because of 2 reasons: (1) the studies differed methodologically, as well as measured multiple different immunological and clinical parameters, (2) some studies had interesting clinical findings, such as the association between duration of HT or thyroid volume and vitamin D deficiency severity. We believe that a description of conclusions from the studies put in the table would be convenient for a reader, especially a clinician.
In Table 2 (now Table 3), we chose to present only the direction of changes because we believe that as the table shows various parameters (which were measured in some studies, and others not), avoiding presenting precise quantitative data would make the table a little more clear to read. There is still little data on the influence of vitamin D supplementation on changes in parameters other than anti-thyroid antibodies, so it was crucial for us to highlight these markers themselves in a separate column.